# Selective light absorber-assisted single nickel atom catalysts for ambient sunlight-driven $CO_2$ methanation

Yaguang Li [1,8], Jianchao Hao[1,8], Hui Song[2,3,8], Fengyu Zhang[1], Xianhua Bai[1], Xianguang Meng[4], Hongyuan Zhang[5], Shufang Wang[1], Yong Hu [5] & Jinhua Ye [2,3,6,7]

Ambient sunlight-driven $CO_2$ methanation cannot be realized due to the temperature being less than 80 °C upon irradiation with dispersed solar energy. In this work, a selective light absorber was used to construct a photothermal system to generate a high temperature (up to 288 °C) under weak solar irradiation (1 kW m$^{-2}$), and this temperature is three times higher than that in traditional photothermal catalysis systems. Moreover, ultrathin amorphous $Y_2O_3$ nanosheets with confined single nickel atoms (SA Ni/$Y_2O_3$) were synthesized, and they exhibited superior $CO_2$ methanation activity. As a result, 80% $CO_2$ conversion efficiency and a $CH_4$ production rate of 7.5 L m$^{-2}$ h$^{-1}$ were achieved through SA Ni/$Y_2O_3$ under solar irradiation (from 0.52 to 0.7 kW m$^{-2}$) when assisted by a selective light absorber, demonstrating that this system can serve as a platform for directly harnessing dispersed solar energy to convert $CO_2$ to valuable chemicals.

[1] Hebei Key Lab of Optic-electronic Information and Materials, The College of Physics Science and Technology, Hebei University, Baoding 071002, P. R. China. [2] Graduate School of Chemical Sciences and Engineering, Hokkaido University, Sapporo 060-0814, Japan. [3] International Center for Materials Nanoarchitectonics (WPI-MANA), National Institute for Materials Science (NIMS), 1-1 Namiki, Tsukuba, Ibaraki 305-0044, Japan. [4] Hebei Provincial Key Laboratory of Inorganic Nonmetallic Materials, College of Materials Science and Engineering, North China University of Science and Technology, Tangshan 063210 Hebei, P. R. China. [5] Department of Chemistry, Zhejiang Normal University, Jinhua, Zhejiang 321004, China. [6] TJU-NIMS International Collaboration Laboratory, School of Material Science and Engineering, Tianjin University, Tianjin 300072, P. R. China. [7] Collaborative Innovation Center of Chemical Science and Engineering (Tianjin), Tianjin 300072, P. R. China. [8] These authors contributed equally: Yaguang Li, Jianchao Hao, Hui Song. Correspondence and requests for materials should be addressed to Y.L. (email: yaguang_1987@126.com) or to S.W. (email: sfwang@hbu.edu.cn) or to Y.H. (email: yonghu@zjnu.edu.cn) or to J.Y. (email: Jinhua.YE@nims.go.jp)

The rapid consumption of fossil fuels has caused serious energy shortage problems, as well as greenhouse effects[1–3]. Methane ($CH_4$) is the main component of natural gas and is widely used as a source of clean energy with low carbon emissions[4,5]. Converting $CO_2$ into synthetic natural gas through methanation has great significance for mitigating $CO_2$ emissions[6,7] and realizing hydrogen storage[8,9], as the excess electric power generated at night can be used for $H_2$ production[9–11]. For $CO_2$ methanation, a temperature of at least 200 °C is needed to activate the catalytic reaction[12,13], thus requiring a secondary energy source[14,15]. Solar-driven $CO_2$ methanation via a photothermal effect represents a promising strategy to produce $CH_4$ without secondary energy input[16–20]. However, intense light irradiation (more than 10 kW m$^{-2}$, equal to ten times the standard intensity of solar light) must be provided to heat the catalysts to 200 °C to drive the $CO_2$ methanation[5,21]. Such intense irradiation requires complex instruments and increased energy consumption, limiting the potential of photothermal catalysis for industrial applications[22]. Thus, employing weak solar light (1 kW m$^{-2}$) to achieve high temperatures for photothermal $CO_2$ methanation as well as other photothermal catalytic reactions is quite challenging. To achieve this goal, metallic nanoparticles, carbon-based materials, etc. have been widely investigated as photothermal catalysts due to their ability to absorb the full solar spectrum[23,24]. Although they can absorb the full solar spectrum, their thermal radiation is high due to their blackbody nature[25,26]. The strong thermal radiation prevents heat storage by photothermal materials, leading to a 100 °C limit of photothermal materials under 1 sun[27,28], which is not suitable for triggering photothermal $CO_2$ methanation.

On the other hand, ruthenium (Ru) catalysts have been verified as the best catalysts for photothermal $CO_2$ methanation[5,20]. Since Ru is scarce, noble metal-free catalysts have to be developed to replace expensive Ru catalysts[29]. Unfortunately, the $CO_2$ methanation reactions catalyzed by conventional base metals are sluggish[12,30]. Therefore, the rational design of efficient methanation catalysts using base metals is highly desired. Recently, supported single-atom catalysts have demonstrated high activity in various reactions, such as hydrogenation, oxidation, and water–gas shift reactions[31–34]. Nickel-based catalysts are among the most active base-metal catalysts for thermal $CO_2$ methanation[35]. However, until now, few studies have been reported on isolated Ni atom catalysts for photothermal $CO_2$ methanation.

In this work, benefiting from the addition of selective light absorbers that can both absorb the full solar spectrum and generate little thermal radiation[36], we fabricate a photothermal conversion device that can heat catalysts to 288 °C under 1 sun (1 kW m$^{-2}$) and successfully propel weak sunlight-driven photothermal $CO_2$ methanation. Furthermore, we prepared two-dimensional amorphous $Y_2O_3$ nanosheets decorated with single Ni atoms (SA Ni/$Y_2O_3$). Compared with other Ni-based catalysts, SA Ni/$Y_2O_3$ nanosheets show a lower initial reaction temperature and higher activity in $CO_2$ methanation. Coupled with the selective light absorber-assisted photothermal system, the SA Ni/$Y_2O_3$ nanosheets exhibit efficient and stable photothermal $CH_4$ production under solar irradiation.

## Results

**Traditional and selective light absorber-assisted photothermal systems**. $Y_2O_3$ nanosheets loaded with 4% wt Ni nanoparticles were prepared to demonstrate the solar absorption and heat storage of the state-of-the-art photothermal catalysts (Ni/$Y_2O_3$, Supplementary Fig. 1, synthesis details can be found in the Methods section). Figure 1a shows the black color of the Ni/$Y_2O_3$ nanosheets. The ultraviolet–visible–infrared (UV–Vis–IR)

absorption spectrum of the Ni/$Y_2O_3$ nanosheets exhibits a photoresponse in the ultraviolet (300 nm) to visible and near-infrared (IR) spectral regions (up to 2 μm), confirming the 100% solar absorption capacity of the Ni/$Y_2O_3$ nanosheets (Fig. 1b). Furthermore, the Ni/$Y_2O_3$ nanosheets exhibit a high level of IR absorption in the 2–10 μm range. Thermal radiation of materials is known as IR radiation[37]. Kirchhoff's law states that the IR emissivity is equal to the IR absorptivity of the material in thermodynamic equilibrium[38]. Therefore, the IR absorption of the Ni/$Y_2O_3$ nanosheets leads to a strong IR radiation by the Ni/$Y_2O_3$ nanosheets, resulting in intense heat dissipation[39]. Assuming that the Ni/$Y_2O_3$ nanosheets are at 200 °C (Fig. 1c), the IR radiative heat loss can be as high as 2.27 kW m$^{-2}$, equivalent to 2.27 standard suns (based on the Stefan–Boltzmann law, see the details in the Methods section), far exceeding the one solar flux of 1 kW m$^{-2}$. This is the reason why the Ni/$Y_2O_3$ nanosheets can only reach 78 °C (Fig. 1h) and the $CO_2$ methanation cannot be conducted with the Ni/$Y_2O_3$ nanosheets under one sun (Fig. 1i). Therefore, high IR radiation is the bottleneck of photothermal materials for obtaining high temperatures under weak solar irradiation[40]. To overcome the energy radiation obstacle, we selected a selective light absorber (AlN$_x$/Al foil) consisting of Al foil with a 2-μm thick coating of AlN$_x$, as shown in Fig. 1d and Supplementary Fig. 2. The UV–Vis–IR absorption spectrum of the selective light absorber exhibits an obvious absorption from 300 nm to 1300 nm (Fig. 1e), accounting for ~100% of the solar spectrum, and little absorption in the IR region of 2–10 μm. According to the simulated calculations (Fig. 1f), the radiative heat loss from the selective light absorber is 0.21 kW m$^{-2}$ at 200 °C (see calculation details in the Methods section), equaling 10% of the thermal radiation of the Ni/$Y_2O_3$ nanosheets at 200 °C and 21% of the standard solar flux. We compared the temperatures of several typical photothermal materials under 1 sun, and 103 °C was the highest temperature reached by photothermal materials with vacuum thermal insulation (Supplementary Table 1), which is 34% of the temperature achieved by a selective light absorber (300 °C). We designed a photothermal system using a selective light absorber. As shown in Fig. 1g and Supplementary Fig. 3, the outside of a quartz tube is coated with the selective light absorber to achieve the high temperature. The Ni/$Y_2O_3$ nanosheets were coated onto the inside of a flow-type quartz tube. Figure 1h indicates that the temperature of the Ni/$Y_2O_3$ nanosheets in the selective light absorber-assisted photothermal system can be as high as 288 °C under 1.0 kW m$^{-2}$ solar irradiation, 3.7 times higher than that of Ni/$Y_2O_3$ nanosheets directly irradiated by the same light. The photothermal $CO_2$ methanation efficiency reached 80% under 1.0 kW m$^{-2}$ solar illumination (Fig. 1i). For comparison, the traditional photothermal system reported by Ye et al. showed that photothermal $CO_2$ methanation could not be achieved by Ni/Al$_2O_3$, due to the one solar-driven temperature of 81 °C[20], revealing the significant advantage of a selective light absorber in weak sunlight-driven photothermal catalysis.

**Preparation and characterization of SA Ni/$Y_2O_3$ nanosheets**. In addition to increasing the photon-induced temperature, we also tried to enhance the activity of Ni-based catalysts. The size of the Ni species can be as small as a single atom on the $Y_2O_3$ nanosheets (SA Ni/$Y_2O_3$) by a bimetal ion-adsorption method[41]. As illustrated in Fig. 2a, briefly, Ni(NO$_3$)$_2$·6H$_2$O and Y(NO$_3$)$_3$·6H$_2$O were dissolved in water to obtain a solution. Then, graphene oxide (GO) was added into the solution to adsorb the Ni$^{2+}$ and Y$^{3+}$ metal ions. The SA Ni/$Y_2O_3$ nanosheets were obtained after the washing, freeze drying, and annealing (see the Methods for synthesis details). As determined by inductively

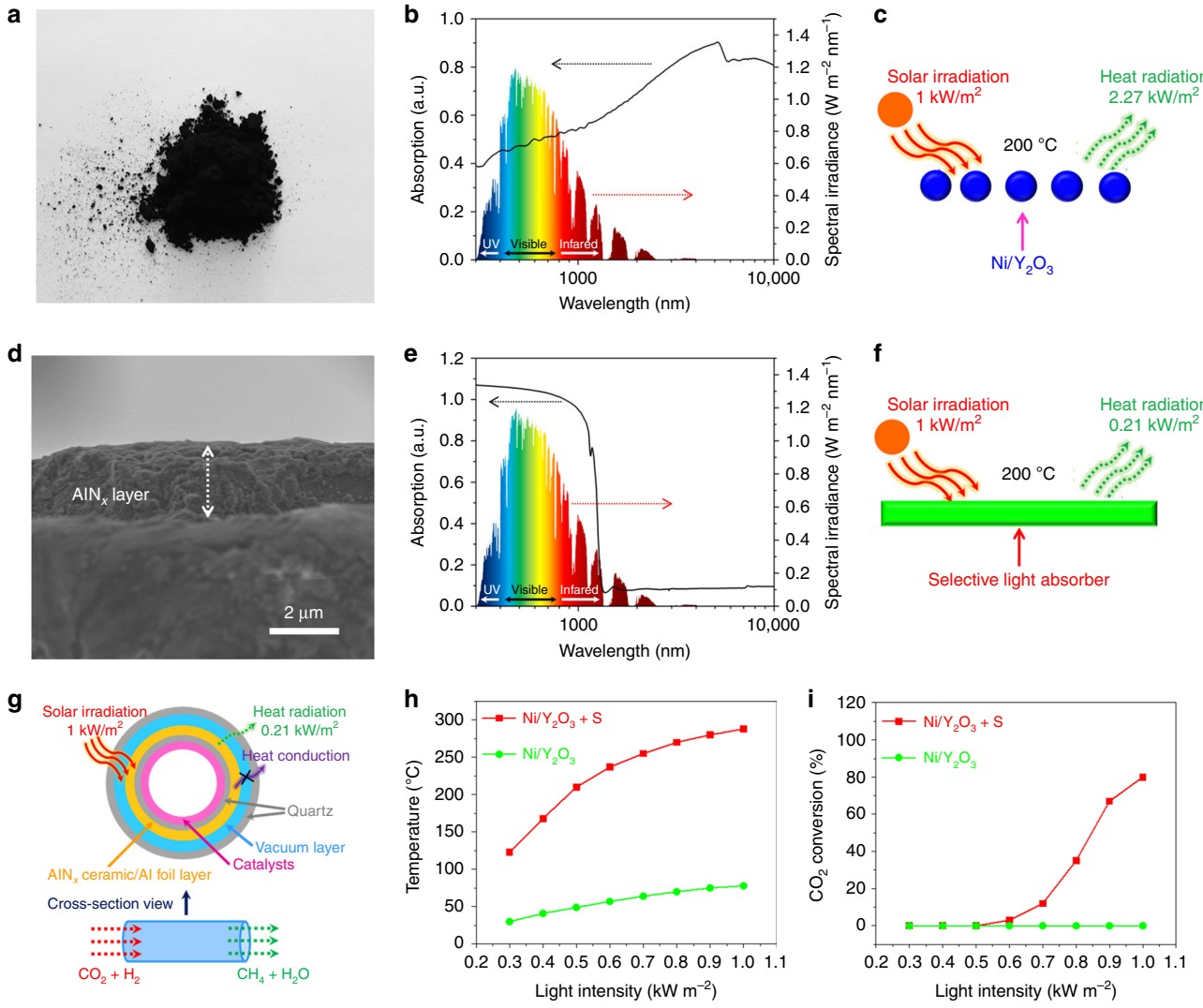

**Fig. 1** Traditional and selective light absorber-assisted photothermal systems. **a** Photograph of the $Ni/Y_2O_3$ nanosheets powder. **b** Normalized UV–Vis–IR absorption spectrum of the $Ni/Y_2O_3$ nanosheets. **c** Solar energy absorption and thermal radiation diagram for the $Ni/Y_2O_3$ nanosheets at 200 °C. The ambient solar flux of $1\,kW\,m^{-2}$ is not enough to sustain the thermal radiation; thus, an equilibrium temperature of 200 °C cannot be obtained. **d** Cross-sectional SEM image and **e** normalized UV–Vis–IR absorption spectrum of the selective light absorber ($AlN_x/Al$ foil). **f** Solar energy absorption and thermal radiation diagram of the selective light absorber at 200 °C. The thermal radiation was far below the absorbed energy from ambient solar flux. **g** Schematic of the new photothermal system used for photothermal $CO_2$ methanation with the selective absorber and the catalysts. **h, i** The light-driven temperature and $CO_2$ conversion rates of the $Ni/Y_2O_3$ nanosheets with ($Ni/Y_2O_3$ + S red) and without ($Ni/Y_2O_3$ green) the selective light absorber-assisted photothermal system, respectively, under different intensities of sunlight irradiation. The scale bar in **d** is 2000 nm

coupled plasma–atomic emission spectroscopy (ICP-AES), the weight percentage of Ni in the SA $Ni/Y_2O_3$ nanosheets was 3.9 wt % (in this paper, the amount of Ni in all the SA $Ni/Y_2O_3$ nanosheets is 3.9 wt%, unless otherwise stated). Figure 2b shows the X-ray diffraction (XRD) pattern of the SA $Ni/Y_2O_3$ nanosheets, indicating the amorphous nature of the product. The Brunauer–Emmett–Teller (BET) specific area of the SA $Ni/Y_2O_3$ nanosheets (Fig. 2c) was as high as $425\,m^2\,g^{-1}$, ensuring a large number of active sites for catalytic reactions. The low-magnification scanning electron microscopy (SEM) image (Fig. 2d) revealed that the SA $Ni/Y_2O_3$ nanosheets exhibit a silk-like morphology with a typical length of a few micrometers. The transmission electron microscopy (TEM) image in Fig. 2e further confirms the ultrathin structure of the 2D nanosheets. The average thickness of the SA $Ni/Y_2O_3$ nanosheets is ~1.5 nm (Supplementary Fig. 4). The high-resolution TEM image (Fig. 2f),

and the corresponding selected area electron diffraction (SAED) pattern (inset in Fig. 2f) reveal no lattice fringes or indistinct diffraction rings in the SA $Ni/Y_2O_3$ nanosheets, confirming the absence of Ni nanoparticles. Moreover, the elemental mapping of the SA $Ni/Y_2O_3$ nanosheets revealed that Ni, Y, and O are homogeneously distributed throughout the SA $Ni/Y_2O_3$ nanosheets (Fig. 2g).

To confirm the presence state of Ni on the SA $Ni/Y_2O_3$ sheets, aberration-corrected TEM measurements were performed. Figure 3a indicates that no precipitates larger than 1 nm can be found on the nanosheets, and only several small dark dots were visible on the nanosheets. These dots were 1–3 Å in diameter, similar to the size of single atoms or small clusters composed of several atoms. To further verify the structure of the Ni species on the SA $Ni/Y_2O_3$ nanosheets, extended X-ray absorption fine structure (EXAFS) analysis was conducted. As shown in Fig. 3b,

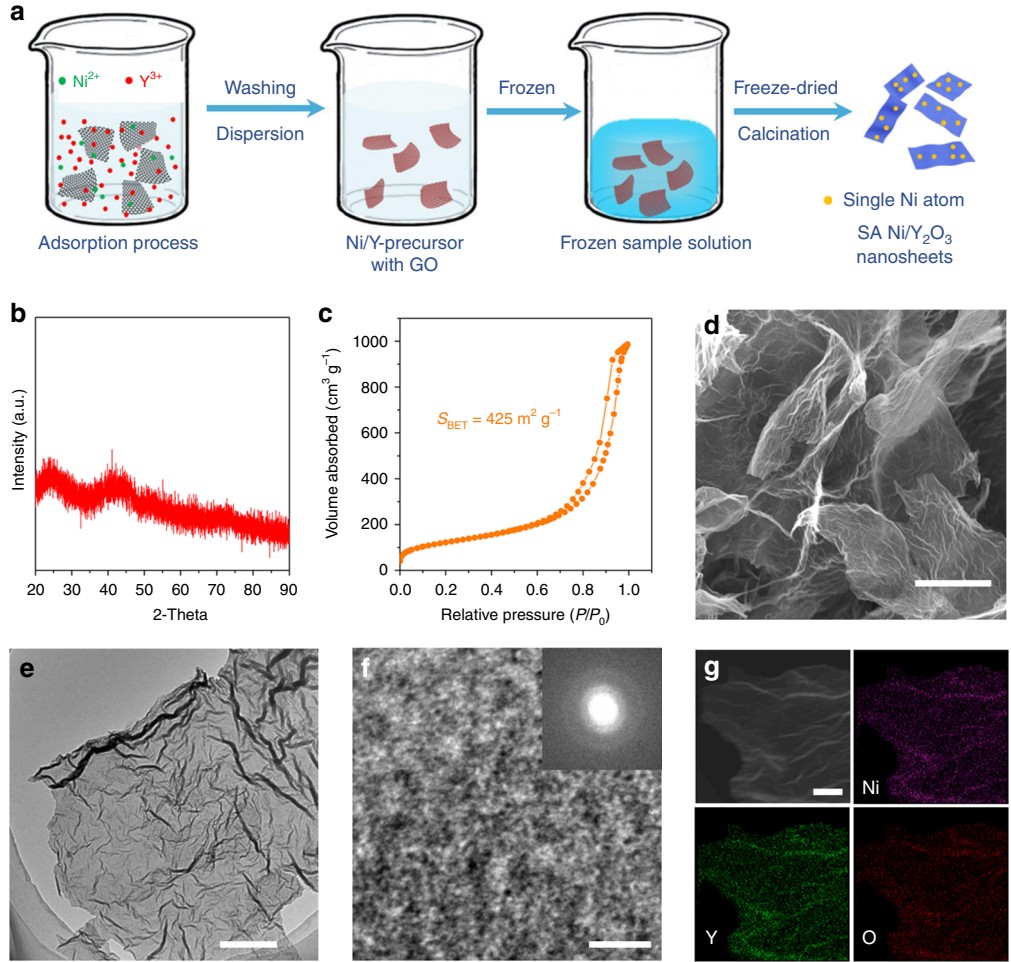

**Fig. 2** Preparation and characterization of the SA Ni/$Y_2O_3$ nanosheets. **a** Schematic of the preparation process for $Y_2O_3$ nanosheets decorated with single Ni atoms (SA Ni/$Y_2O_3$). **b** XRD pattern, **c** $N_2$ adsorption–desorption isotherm, **d** SEM image, **e** TEM image, **f** HRTEM image, **g** STEM image, and EDS mapping images of Ni, Y, and O of the SA Ni/$Y_2O_3$ nanosheets. The inset in **f** is the corresponding electron diffraction pattern. The scale bars in **d**, **e**, **f**, and **g** are 1000, 200, 2, and 20 nm, respectively

the EXAFS curves show that the near-edge absorption energy of the SA Ni/$Y_2O_3$ nanosheets is higher than that of Ni foil and lower than that of NiO, showing that the oxidation state of the Ni species in Ni/$Y_2O_3$ nanosheets (Supplementary Fig. 5). The Fourier transform (FT)-EXAFS curve (Fig. 3c) for the SA Ni/$Y_2O_3$ nanosheets shows only a main peak at ~1.7 Å, which could be attributed to Ni–O scattering. The small signals from Ni–Ni at approximately 2.1 Å and Ni–O–Ni at ~2.5 Å[42,43] confirm that the Ni is mainly distributed as single atoms on the SA Ni/$Y_2O_3$ nanosheets. The structure of amorphous $Y_2O_3$ decorated with single Ni atoms is illustrated in the inset of Fig. 3d. The simulated Ni-coordinated FT-EXAFS spectrum of this model (Fig. 3d) fits well with the measured spectrum of the SA Ni/$Y_2O_3$ nanosheets, further confirming single Ni atoms are the dominant Ni species on the SA Ni/$Y_2O_3$ nanosheets.

**$CO_2$ methanation performance of the SA Ni/$Y_2O_3$ nanosheets**. The catalytic performance of the as-obtained SA Ni/$Y_2O_3$ nanosheets was evaluated in $CO_2$ hydrogenation. First, an experiment was performed using the $Y_2O_3$ nanosheets as the catalyst. The $CO_2$ conversion by the $Y_2O_3$ nanosheets was negligible, as presented in Supplementary Fig. 6. With the SA Ni/$Y_2O_3$ nanosheets, $CO_2$ reduction started at ~180 °C, and a conversion rate of ~87% was achieved at 240 °C (Fig. 4a), while the

$CO_2$ conversion with Ni/$Y_2O_3$ nanosheets was ~11% at the same temperature, corresponding to 13% of that achieved with the SA Ni/$Y_2O_3$ nanosheets. Since the amount of Ni and the surface areas of the SA Ni/$Y_2O_3$ nanosheets and Ni/$Y_2O_3$ nanosheets are similar (Supplementary Table 2), the difference in the $CO_2$ conversion reveals that the Ni single atoms lead to the high $CO_2$ methanation activity of the SA Ni/$Y_2O_3$ nanosheets in comparison with the Ni/$Y_2O_3$ nanosheets. We have compared some noble metal-free catalysts for thermal $CO_2$ methanation in Supplementary Table 3. The thermal $CO_2$ methanation activity of the SA Ni/$Y_2O_3$ nanosheets is higher than those reported for other Ni-based catalysts[13,44–46]. Figure 4b shows that nearly 100% selectivity for $CH_4$ formation is achieved in the thermal $CO_2$ hydrogenation with the SA Ni/$Y_2O_3$ nanosheets at different reaction temperatures, revealing high selectivity for $CH_4$. During the 90-h heating/cooling tests, the $CO_2$ conversion rate of the SA Ni/$Y_2O_3$ nanosheets at 240 °C was maintained at ~87% (Fig. 4c), indicating excellent catalytic stability. Supplementary Fig. 7 shows that the morphology of the SA Ni/$Y_2O_3$ nanosheets was retained after 90 h of testing, and several small dark dots remained on the SA Ni/$Y_2O_3$ nanosheets after the stability test, but cleat Ni precipitation was not observed (Fig. 4d), confirming the robustness of the SA Ni/$Y_2O_3$ nanosheets in thermal $CO_2$ methanation.

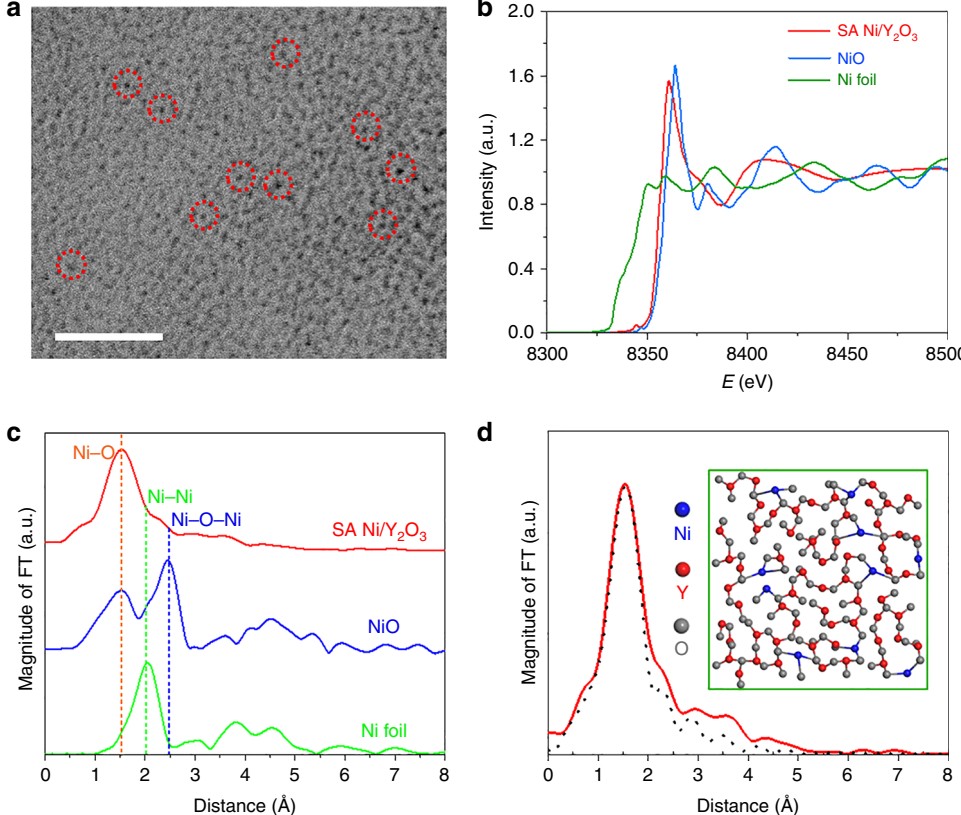

**Fig. 3** Characterization of the Ni in the SA Ni/Y$_2$O$_3$ nanosheets. **a** Aberration-corrected TEM image of the SA Ni/Y$_2$O$_3$ nanosheets. **b** EXAFS spectra of the Ni K-edge of the SA Ni/Y$_2$O$_3$ nanosheets, NiO and Ni foil. **c** Fourier transform (FT) of the Ni K-edge of the SA Ni/Y$_2$O$_3$ nanosheets, NiO and Ni foil. **d** Schematic model of the SA Ni/Y$_2$O$_3$ nanosheets and the corresponding FT-EXAFS fitting curves for the SA Ni/Y$_2$O$_3$ nanosheets. The scale bar in **a** is 2 nm

The photothermal CO$_2$ methanation with the SA Ni/Y$_2$O$_3$ nanosheets assisted by a selective light absorber is depicted in Fig. 5. Figure 5a shows the spatial temperature mapping of the photothermal system under simulated 1 sun (1.0 kW m$^{-2}$) in air. The highest temperature recorded by the IR camera was ~255 °C. When the gas inlet and outlet are encapsulated by thermal insulating covers, the temperature of the SA Ni/Y$_2$O$_3$ nanosheet layer probed by the thermometer increased to 285 °C under 1.0 kW m$^{-2}$ solar irradiation and exceeded 200 °C under 0.5 kW m$^{-2}$ irradiation. The photothermal CO$_2$ methanation started at only 0.4 kW m$^{-2}$ irradiation, and the conversion rate was 90% under one sun (Fig. 5b). The selective sunlight absorber does not affect the high CH$_4$ selectivity of the SA Ni/Y$_2$O$_3$ nanosheets, and nearly 100% CH$_4$ selectivity was observed under different intensities of light irradiation (Supplementary Fig. 8). Moreover, the SA Ni/Y$_2$O$_3$ nanosheets also show excellent stability in the photothermal system under 1 sun (Supplementary Fig. 9). We directly performed CO$_2$ methanation experiments under outdoor sunlight. The experiment was carried out from 08:00 to 18:00 under natural sunlight with a maximum intensity of ~0.7 kW m$^{-2}$ (Fig. 5c). As displayed in Fig. 5d, the CO$_2$ methanation started at 9:00 am and increased gradually as time passed. The conversion remained at ~90% from 11:00 to 16:00 and decreased after 16:00. We calculated the photothermal CO$_2$ methanation rate from 10:00 to 16:00. The CH$_4$ production rate was 7.5 L m$^{-2}$ h$^{-1}$ under ambient sunlight (the calculation details are provided in the Methods section), indicating that industrial-grade CO$_2$ methanation has been achieved under ambient sunlight without additional energy input.

## Discussion

In this study, we have synthesized amorphous Y$_2$O$_3$ nanosheets decorated with single Ni atoms at a 3.9% Ni mass ratio (SA Ni/Y$_2$O$_3$ nanosheets), having a specific surface area of 425 m$^2$ g$^{-1}$. The SA Ni/Y$_2$O$_3$ nanosheets exhibit a lower initiating temperature of 180 °C for CO$_2$ methanation, ~100% methanation selectivity and ~87% CO$_2$ conversion at 240 °C. In particular, a selective light absorber was employed to construct the photothermal system, which can reach 288 °C under one sun. Therefore, the solar-driven photothermal CO$_2$ methanation using the SA Ni/Y$_2$O$_3$ nanosheets exhibits 90% conversion efficiency and 100% methane selectivity with the assistance of a selective light absorber. Moreover, under ambient daytime sunlight (from 0.52 to 0.7 kW m$^{-2}$), a photothermal CO$_2$ methanation conversion efficiency of more than 80% and a methane production rate of 7.5 L mol m$^{-2}$ h$^{-1}$ are achieved using this system. The strategy of coupling an efficient 2D base-metal catalyst with a selective light absorber demonstrates great potential for solar-to-chemical energy conversion.

## Methods

**Chemicals**. Commercial nickel nitrate hexahydrate (Ni(NO$_3$)$_2$·6H$_2$O) and yttrium nitrate hexahydrate (Y(NO$_3$)$_3$·6H$_2$O) were purchased from Kermel Co., Ltd. Graphene oxide (GO) was purchased from Hangzhou Gaoxi Technology Co., Ltd. Liquid nitrogen was purchased from Xicheng Special Gas Co., Ltd. The selective light absorber (AlN$_x$/Al foil) was provided by Hangzhou Ruijia Precision Science Instrument Co., Ltd. All chemicals were directly used as received without further treatment.

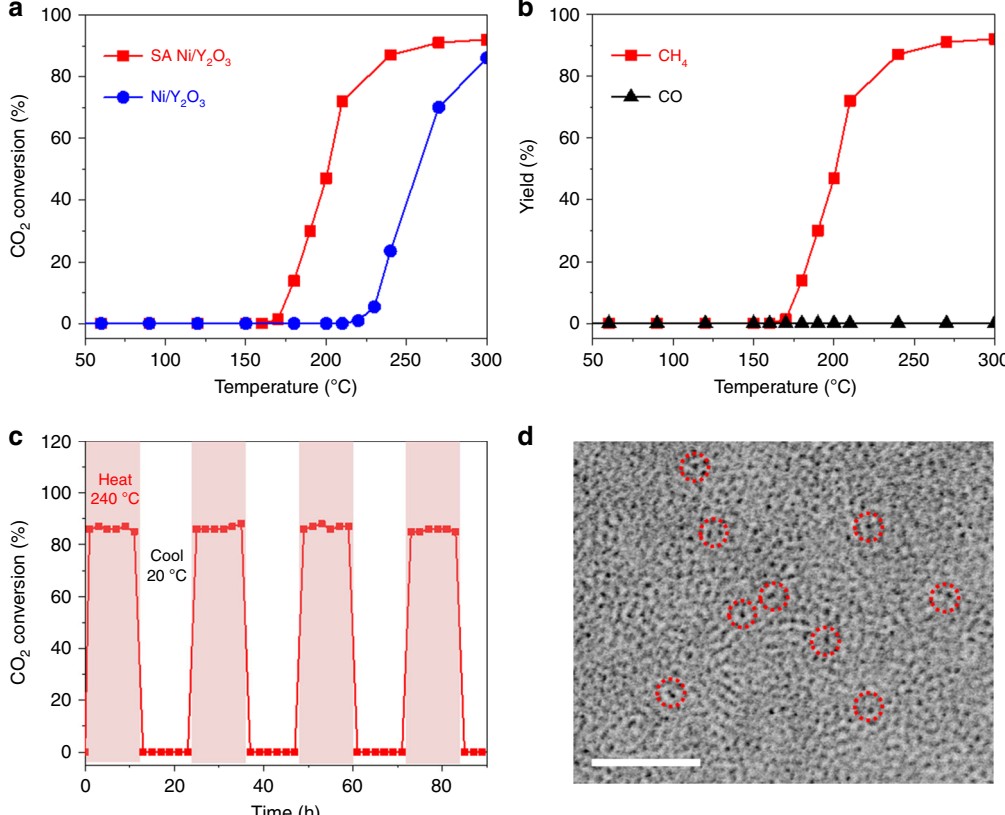

**Fig. 4** Thermocatalytic $CO_2$ hydrogenation experiments. **a** Thermal $CO_2$ conversion using the SA $Ni/Y_2O_3$ nanosheets (SA $Ni/Y_2O_3$) and Ni nanoparticles/$Y_2O_3$ nanosheets ($Ni/Y_2O_3$) as a function of temperature. **b** $CH_4$ and CO yields from the $CO_2$ hydrogenation over the SA $Ni/Y_2O_3$ nanosheets as a function of temperature. **c** $CO_2$ hydrogenation versus reaction time over the SA $Ni/Y_2O_3$ nanosheets at 240 °C. **d** Aberration-corrected TEM image of the SA $Ni/Y_2O_3$ nanosheets after the stability test shown in Fig. 4c. Reaction conditions: 100 ml min$^{-1}$ of reaction gas (2.5% $CO_2$ + 10% $H_2$ + 87.5% $N_2$), 100 mg of catalyst. The scale bar in **d** is 2 nm

**Synthesis of $Ni/Y_2O_3$ nanosheets**. To synthesize the $Ni/Y_2O_3$ nanosheets, we first synthesized $Y_2O_3$ nanosheets. Three grams of $Y(NO_3)_3$ was dispersed in 100 ml of water, and then 100 ml of GO dispersed in water (2 mg/ml) was added into the above solution to adsorb the $Y^{3+}$ metal ions on the surface of the GO sheets. After washing and centrifugation with water, the product was dispersed in pure deionized water by ultrasonic treatment. Subsequently, the solution was frozen in liquid nitrogen and then freeze-dried for 3 days. The product was calcined in air at 400 °C for 24 h to remove the GO nanosheets and mineralize the metal ions as $Y_2O_3$ nanosheets. Then, a certain amount of $Ni(NO_3)_2$ was dissolved in water. After that, 200 mg of the $Y_2O_3$ nanosheets was dispersed into the above solution. After stirring for 2 h, the samples were dried at 80 °C and calcined in air at 400 °C for 4 h to prepare the NiO-loaded $Y_2O_3$ nanosheets ($NiO/Y_2O_3$ nanosheets). Then, the $NiO/Y_2O_3$ nanosheets were reduced under 10% $H_2/Ar$ at 400 °C for 2.5 h to prepare the $Ni/Y_2O_3$ nanosheets. The loadings of Ni were ~4.0 wt% based on ICP-AES.

**Synthesis SA $Ni/Y_2O_3$**. To synthesize 3.9 wt% SA $Ni/Y_2O_3$ nanosheets, we selected single-layered exfoliated GO sheets as the substrate. First, 0.8 g of Ni $(NO_3)_2$ and 3 g of $Y(NO_3)_3$ were dispersed in 100 ml of water, and then 100 ml of GO dispersed in water (2 mg/ml) was added into the above solution to adsorb the $Ni^{2+}$ and $Y^{3+}$ metal ions onto the surface of the GO sheets. After washing with water and centrifugation, the product was dispersed in pure deionized water by ultrasonic treatment. Subsequently, the solution was frozen in liquid nitrogen and then freeze-dried for 3 days. The product was calcined in air at 400 °C for 24 h to remove the GO nanosheets and mineralize the metal ions as metal oxide nanosheets. Then, the 2D metal oxide nanosheets were reduced with 10% $H_2/Ar$ at 400 °C for 2.5 h to obtain the SA $Ni/Y_2O_3$ nanosheets. The loadings of Ni were ~3.9 wt% based on ICP-AES.

**Catalyst characterization**. Scanning electron microscopy (FEI Nova Nano SEM450) was used to identify the holistic and porous morphologies of the samples. The prepared samples were studied by powder X-ray diffraction, which was performed on a Bede D1 system operated at 20 kV and 30 mA with Cu Kα radiation

(λ = 1.5406 Å). TEM (JEOL 2100 plus + ARM 200 F) was used to identify the morphology and crystal structure of the nanostructures as well as EDS mapping. The XPS spectra were recorded on a Thermo ESCALAB-250 spectrometer using a monochromatic Al Kα radiation source (1486.6 eV). The binding energies determined by XPS were corrected by referencing the adventitious carbon peak (284.6 eV) for each sample. The BET surface areas were obtained using a Micromeritics Tristar 3020 system.

**EXAFS test and analysis**. The ex situ Ni K-edge extended X-ray absorption fine structure (EXAFS) data were collected on the beamline at Shanghai Synchrotron Radiation Facility (SSRF). All samples were prepared by placing a small amount of homogenized (via agate mortar and pestle) powder on 3 M tape. We used IFEFFIT software to calibrate the energy scale, correct the background signal and normalize the intensity. The spectra were normalized with respect to the edge height after subtracting the pre-edge and post-edge backgrounds using Athena software. To extract the EXAFS oscillations, the background was removed in k-space using a five-domain cubic spline. The resulting k-space data were then Fourier transformed.

**Thermal $CO_2$ methanation test**. First, 100 mg of the catalyst was transferred to the reaction cell and pre-reduced at 400 °C for 2.5 h under $H_2$ gas (10% $H_2$ + 90% Ar) flowing at 100 ml min$^{-1}$. After cooling to room temperature, the reaction gas, a mixture of 2.5% $CO_2$ + 10% $H_2$ + 87.5% $N_2$, was introduced. The catalyst was heated to the desired reaction temperature at a heating rate of 5 °C min$^{-1}$. The temperature was detected using a platinum resistance thermometer (M363886). The catalyst was kept at each temperature for 30 min. Then, the composition of the outlet gas was analyzed by an on-line gas chromatograph equipped with a flame ionization detector (FID).

When calculating the turnover frequencies (TOF values), we assumed that all the Ni in the SA $Ni/Y_2O_3$ took part in $CO_2$ methanation. The TOF value for the methanation of $CO_2$ was calculated based on the formula below:

$$\text{TOF} = \left(V_{CO_2}^* A^* 59\right) / \left(Ni_a^* 22.4\right) \tag{1}$$

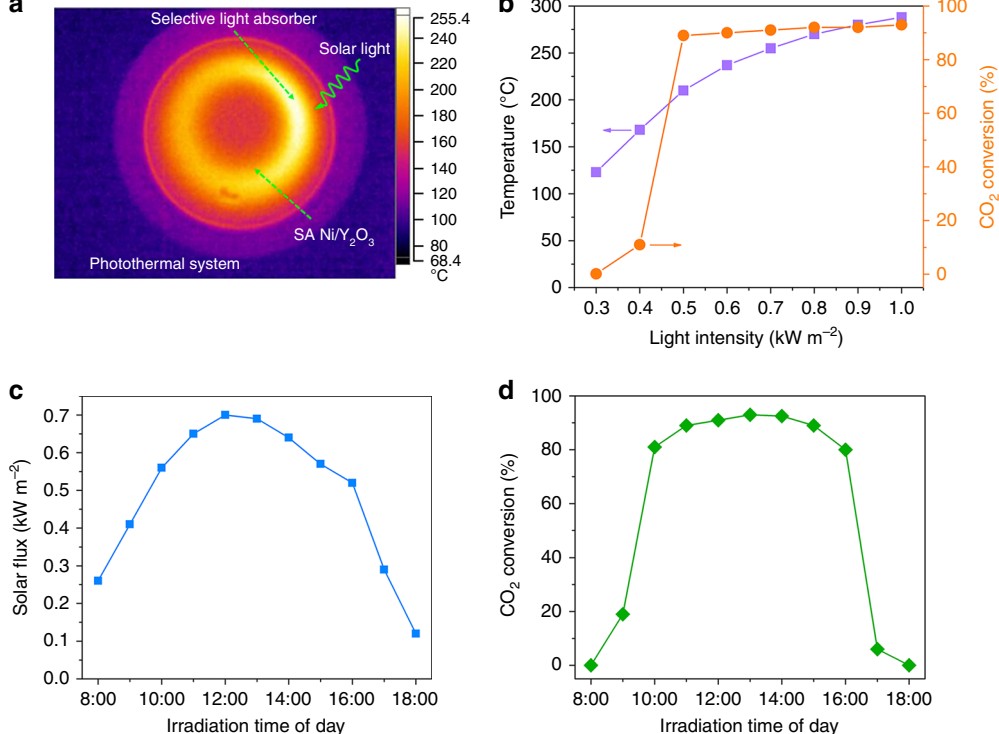

**Fig. 5** Photothermal $CO_2$ methanation performance of the SA $Ni/Y_2O_3$ nanosheets with a selective light absorber-assisted photothermal system. **a** Spatial temperature mapping of the selective light absorber-assisted quartz tube coated with the SA $Ni/Y_2O_3$ nanosheets under 1.0 kW m$^{-2}$ of simulated solar irradiation obtained by an infrared camera. **b** The temperature and $CO_2$ conversion achieved by the SA $Ni/Y_2O_3$ nanosheets with the selective light absorber-assisted photothermal system under different intensities of simulated solar light. **c** The solar flux over time on June 30, 2018 from 8:00 to 18:00 in Baoding, Hebei, China. **d** The corresponding photothermal $CO_2$ conversion over the SA $Ni/Y_2O_3$ nanosheets with the selective light absorber-assisted photothermal system as a function of time. Reaction conditions: 100 ml min$^{-1}$ of reaction gas (2.5% $CO_2$ + 10% $H_2$ + 87.5% $N_2$), 100 mg of catalyst

where $V_{CO_2}$ is the $CO_2$ flux (0.042 ml·S$^{-1}$), A is the $CO_2$ conversion rate, and $Ni_a$ is the quantity of nickel in SA $Ni/Y_2O_3$ (3.9 mg).

**Optical property**. The UV–Vis–IR optical properties of the $Ni/Y_2O_3$ nanosheets were investigated on a Shimadzu UV3600 UV–Vis spectrophotometer from 300 to 2000 nm. The absorptions in the IR region (1.5–10 μm) were detected by an FTIR spectrometer (Bruker, VERTEX 70 FT-IR). The UV–Vis–IR optical properties of the selective light absorber (AlN$_x$/Al foil) were provided by Hangzhou Ruijia Precision Science Instrument Co., Ltd.

**Temperature detection**. The temperatures of all materials were determined by a platinum resistance thermometer (M363886).

**Emissivity ($\sigma$) test**. IR photographs were taken with a Fluke Ti300 IR camera (America). The emissivity of the $Ni/Y_2O_3$ nanosheets was tested by an infrared (IR) camera as follows:

First, we heated the sample to the given temperature in a dark environment and measured the temperature of the material with a platinum resistance thermometer. Then, we used an IR camera to measure the temperature of the materials. We changed the emissivity of the IR camera to ensure that the IR temperature was equal to the temperature shown by the thermometer. The emissivity was corrected by factors of 0.93 and 0.95 for the $Ni/Y_2O_3$ nanosheets at 35 and 200 °C, respectively.

The emissivities of the selective light absorber (AlN$_x$/Al foil) were 0.05 and 0.1 at 35 and 200 °C, respectively, which were obtained from the supplier (Hangzhou Ruijia Precision Science Instrument Co., Ltd.).

**Light source with selective light absorber**. The laboratory light source was an array consisting of six xenon lamps (provided by Hebei Scientist Research Experimental and Equipment Trade Co., Ltd.) equipped with an AM 1.5 -G filter to ensure irradiation covering the whole photothermal system constructed by the selective light absorber, and the light intensity can be tuned by the output current. A foam sheet wrapped in aluminium foil was used to support the tube in the photothermal tests. The light intensity was detected by an irradiance meter (I400).

**Light source without selective light absorber**. A xenon lamp (Microsolar 300 from PerfectLight) was used to provide 1 sun intensity for the $Ni/Y_2O_3$ nanosheets. One sun intensity was detected by an irradiance meter (I400).

**Calculation of the thermal radiation**. The thermal radiation was calculated according to the Stefan–Boltzmann law:

$$J = \varepsilon\sigma(T_1^4 - T_2^4) \qquad (2)$$

$J$ is the thermal radiation energy, $\sigma$ is the emissivity of the material, and $T_1$ and $T_2$ are 473 K (200 °C) and 308 K (35 °C), respectively. According to the $\sigma$ of the $Ni/Y_2O_3$ nanosheets and the selective light absorber at different temperatures, the thermal radiation generated by the $Ni/Y_2O_3$ nanosheets and the selective light absorber were 2.27 and 0.21 kW m$^{-2}$, respectively.

**Photothermal catalysis tests**. The photothermal test of 100 mg of catalyst was conducted as follows: 100 mg of catalyst was transferred to a quartz tube reactor, and then a gaseous mixture of 10% $H_2$–90% Ar (100 ml min$^{-1}$) was fed into the reactor. The temperature was raised to 400 °C for 2.5 h. After cooling to room temperature under $N_2$, the reaction gas was fed into the reaction vessel at a rate of 100 sccm. A gaseous mixture of 2.5% $CO_2$–10% $H_2$–87.5% $N_2$ was supplied as the feeding gas. A xenon lamp (Microsolar 300) equipped with an AM 1.5 A G filter was used as the light source. The composition of the outlet gas was analyzed by an on-line gas chromatograph equipped with a flame ionization detector.

**Photothermal catalysis test with selective light absorber**. To replace the electric heating system with the photothermal system, the reaction tube was coated with the sunlight-selective absorber (AlN$_x$/Al foil), as shown in Fig. 1g and Supplementary Fig. 3, and this system was used as the light absorber to achieve a high temperature. One hundred milligrams of the sample was used to coat the inner wall of the tube, and then a gaseous mixture of 10% $H_2$–90% Ar (100 ml min$^{-1}$) was fed into the reactor. Then, the tube was irradiated with 2 suns (2 kW m$^{-1}$) to achieve a high temperature for 2.5 h. After cooling to room temperature, the reaction gas, a mixture of 2.5% $CO_2$ + 10% $H_2$ + 87.5% $N_2$, was introduced. The composition of the outlet gas was analyzed by an on-line gas chromatograph equipped with a flame ionization detector.

**Outdoor catalysis tests**. The outdoor tests were similar to the photothermal catalysis tests. The only difference was that ambient sunlight was the light source. The test day was June 30, 2018.

The $CO_2$ methanation rate ($\eta$) of 7.5 L m$^{-2}$ h$^{-1}$ was calculated by the following formula:

$$\eta\left(Lm^{-2}h^{-1}\right) = K * L * S * 60/0.018 \qquad (3)$$

$K$ (= 87.9%) was the average $CO_2$ conversion efficiency from 10:00 to 16:00 (shown in Fig. 5a), $L$ was the gas flow rate (0.1 L min$^{-1}$), $S$ was the ratio of $CO_2$ in the feed gas (2.10%), 60 was the time (60 min), and 0.018 m$^2$ was the measured irradiation area.

## Data availability

The data that support the findings of this study are available from the corresponding authors upon reasonable requests.

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

## Acknowledgements

This work is supported by the National Nature Science Foundation of China (Grant No. 51702078, 21633004), Outstanding Youth Foundation of Hebei Province (Grant No. A2016201176), Outstanding Doctoral Cultivation Project of Hebei University (Grant No. YB201502), received partial financial support from JSPS KAKENHI Grant Number

JP18H02065, Photo-excitonix Project in Hokkaido University and State Scholarship Fund by China Scholarship Council (CSC) (No. 201606320239). We acknowledge the Shanghai Synchrotron Radiation Facility (SSRF) for conducting the EXAFS experiment (BL14W1).

## Author contributions

Y.L., S.W., Y.H., and J.Y. conceived the project and contributed to the design of the experiments and analysis of the data. Y.L. and H.S. performed the selective light absorber-assisted photothermal system preparation, characterizations. J.H., F.Z. performed the catalyst preparation, characterizations. Y.L., H.S., J.H., X.M. wrote the paper. X.B., H.Z. conducted the SEM and TEM examinations. All the authors discussed the results and commented on the paper.

## Additional information

**Competing interests:** The authors declare no competing interests.

