## [Peer Review File · Nature Communications]

Reviewers' comments:

Reviewer #1 (Remarks to the Author):

The manuscript describes using nickel on yttrium oxide nanosheet catalysts in a solar thermal reactor system for the Sabatier reaction. Under outdoor sunlight conditions as the solar thermal reactor can reach temperatures of 280C and when the nickel yttrium is prepared using a graphene oxide (GO) templating approach the system can produce methane at a rate of 7.5L/m²/hr. The key result from the study is the capacity to drive the Sabatier reaction under ambient outdoor solar conditions without the requirement of a noble metal catalyst. Yttrium oxide (47USD/kg) is more expensive than other common catalyst supports (e.g. alumina (0.4USD/kg)) and fabricating it in nanosheet form will add to that expense although it is likely expected to still be cheaper than utilising a noble metal catalyst for the reaction.

The authors examine two versions of the Ni/Y₂O₃ catalyst, one where the Ni is deposited the Y₂O₃ nanosheets that have been prepared by the GO templating method and a second where the Ni is included with the Y₂O₃ during the GO templating method. Catalyst produced by the second approach gives a much higher activity for the Sabatier reaction under the outdoor operating environment. The authors claim the Ni is dispersed as single atoms (SA) within the Y₂O₃ nanosheet which is what invokes the better performance. The Ni loading on the SA Ni/Y₂O₃ is 3.9wt% which is high for SA systems and the authors use EXAFS (Figure 4b) to show there are no Ni-Ni bonds in the material to support the claim they have a single (Ni) atom catalyst. The EXAFS provides evidence of Ni-O bonds in the material (on comparison with a NiO standard) which suggests the presence of nickel oxide and perhaps indicates highly dispersed Ni throughout the Y₂O₃ but alone does not convincingly indicate SA Ni on the Y₂O₃ nanosheets. Considering Figure 4b, and while subjective, it is possible a peak exists at the Ni-Ni distance for the SA Ni/Y₂O₃. The authors should model the EXAFS spectra for the proposed schematic in Figure 4d to assist with giving evidence of the SA Ni presence.

In relation to the SA Ni, AFM is used to determine the thickness of the nanosheets and in the associated image (SI Fig 5a) there is the presence of regular white spots on the SA Ni/Y₂O₃ nanosheets – are these NiO deposits or some other artefact of the imaging process. The authors need more convincing evidence that the Ni is present as SA than what is currently provided.

Additional characterisation of the Ni/Y₂O₃ catalysts (both the deposited and 'SA' versions) is needed (and could assist with identifying the presence of SA Ni). At a minimum, assessing Ni reducibility, Ni dispersion and high resolution imaging of the catalysts should be performed.

Measuring Ni dispersion would enable TOF to be determined and be a better comparison to illustrate the strong activity of SA Ni/Y₂O₃ (and its origin) relative to the other systems listed in Table S2.

The authors use 12hrs of continuous operation under simulated solar conditions to demonstrate catalyst stability and run it over one outdoor cycle to demonstrate stable performance which is insufficient to support the claim of a stable catalyst. The catalyst needs to be run for a much longer time-frame as well as be subject to multiple heating/cooling cycles (to replicate outdoor operation) before this claim can be made with conviction.

Additionally, the catalyst should be characterised post-stability and post-cycling tests to ensure it has maintained its original integrity.

According to Figure 5b the SA Ni/Y₂O₃ catalyst gives a methane yield of around 90% with no CO produced. Does this mean that 10% of the carbon is unaccounted for or that the 90% yield is a reflection of the 90% CO₂ conversion reached by the system? If it is the later, then the result would be better presented as a selectivity % rather than a yield %. If it is the former, then what has happened to the other 10% of carbon?

The written English requires substantial improvement to remove the many grammatical errors and spelling mistakes (e.g. temperature, anchored, irradiation). Words such as 'huge' and 'remarkable' should be avoided and some of the terminology (e.g. 'very few', 'is hard to be', 'nearly 100%', 'We listed some excellent catalysts') should be revised to be more scientific. The sentence 'In other words.....materials up to now' doesn't make sense.

While the key finding (solar thermal methane production in an outdoor environment) is of interest, the material characterisation is not sufficiently comprehensive and greater examination of the stability of the material is needed before the work can be considered publishable.

Recommendation: Major revision

Reviewer #2 (Remarks to the Author):

This manuscript reports CO₂ hydrogenation over Ni-based catalyst with heating of solar light. However, hydrogenation of CO₂ over Ni-based catalysts is widely investigated, and it is a commercialized process. Furthermore, it is a common way to heat catalysts by solar light. Increase in heating efficiency by insulating a reactor is a conventional techniques. Therefore, the novelty of this work is not high enough for its publication in Nature Communications. Furthermore, the characterization of catalytic active sites before and after reaction are little. I'm unable to recommend it for publication.

Reviewers' comments

Reviewer #1 (Remarks to the Author):

The manuscript describes using nickel on yttrium oxide nanosheet catalysts in a solar thermal reactor system for the Sabatier reaction. Under outdoor sunlight conditions as the solar thermal reactor can reach temperatures of 280C and when the nickel yttrium is prepared using a graphene oxide (GO) templating approach the system can produce methane at a rate of 7.5L/m²/hr. The key result from the study is the capacity to drive the Sabatier reaction under ambient outdoor solar conditions without the requirement of a noble metal catalyst. Yttrium oxide (47USD/kg) is more expensive than other common catalyst supports (e.g. alumina (0.4USD/kg)) and fabricating it in nanosheet form will add to that expense although it is likely expected to still be cheaper than utilising a noble metal catalyst for the reaction.

Response: We are grateful to the reviewer for the valuable comments. The price of noble metals are far expensive than that of yttrium oxide. For instance, ruthenium is 11567 USD/kg. In this case, we chose the single Ni decorated Y₂O₃ nanosheet as model catalyst to show that single atoms combined with two dimensional (2D) supports could increase the CO₂ methanation activity of noble-metal-free catalysts. Based on this strategy, we are working on exploring other economical 2D supports to replace Y₂O₃ nanosheets in the hope of achieving better results and we will certainly carry out this study in the near future. Thanks!

The authors examine two versions of the Ni/Y₂O₃ catalyst, one where the Ni is deposited on the Y₂O₃ nanosheets that have been prepared by the GO templating method and a second where the Ni is included with the Y₂O₃ during the GO templating method. Catalyst produced by the second approach gives a much higher activity for the Sabatier reaction under the outdoor operating environment. The authors claim the Ni is dispersed as single atoms (SA) within the Y₂O₃ nanosheet which is what invokes the better performance. The Ni loading on the SA Ni/Y₂O₃ is 3.9wt% which is high

for SA systems and the authors use EXAFS (Figure 4b) to show there are no Ni-Ni bonds in the material to support the claim they have a single (Ni) atom catalyst. The EXAFS provides evidence of Ni-O bonds in the material (on comparison with a NiO standard) which suggests the presence of nickel oxide and perhaps indicates highly dispersed Ni throughout the Y_2O_3 but alone does not convincingly indicate SA Ni on the Y_2O_3 nanosheets. Considering Figure 4b, and while subjective, it is possible a peak exists at the Ni-Ni distance for the SA Ni/ Y_2O_3 . The authors should model the EXAFS spectra for the proposed schematic in Figure 4d to assist with giving evidence of the SA Ni presence.

Response: We are grateful for the reviewer's positive and insightful comments. According to the reviewer's comments, we have added the FT-EXAFS fitting curve of our proposed model for SA Ni/ Y_2O_3 in Fig. 1d (Fig. 3d in revised manuscript, see Page 17), and the detailed description was shown in our revised manuscript as below in red color in the text (see Page 8):

Fig. 1 Characterization of the Ni in SA Ni/ Y_2O_3 nanosheets. (a) Aberration-corrected TEM image of SA Ni/ Y_2O_3 nanosheets. (b) EXAFS spectra of the Ni K-edge of SA

Ni/Y₂O₃ nanosheets, NiO and Ni foil. (c) Fourier transform (FT) of the Ni K-edge of SA Ni/Y₂O₃ nanosheets, NiO and Ni foil. (d) Schematic model of SA Ni/Y₂O₃ nanosheet, Ni (blue), Y (red), O (gray) and corresponding FT-EXAFS fitting curves for SA Ni/Y₂O₃ nanosheet. The scale bar in (a) is 2 nm.

“The structure of amorphous Y₂O₃ decorated with single Ni atoms is illustrated in the inset of Fig. 3d. And the simulated Ni coordinated FT-EXAFS spectrum of this model (Fig. 3d) fits with the measured result of the SA Ni/Y₂O₃ nanosheets, further confirming the dominant distribution of single Ni atoms on the SA Ni/Y₂O₃ nanosheets.”

In relation to the SA Ni, AFM is used to determine the thickness of the nanosheets and in the associated image (SI Fig 5a) there is the presence of regular white spots on the SA Ni/Y₂O₃ nanosheets – are these NiO deposits or some other artefact of the imaging process.

Response: We thank the reviewer for pointing this out. The white spots are the fragments crushed from SA Ni/Y₂O₃ nanosheets not the Ni precipitations. In the report in *2D Mater.* **4** (2017) 025031, the AFM images also show that there are also many white spots distributed on the CaO, TiO₂ ultrathin nanosheets, which are caused by the violent ultrasonication during the preparation of samples for AFM test.

The authors need more convincing evidence that the Ni is present as SA than what is currently provided. Additional characterization of the Ni/Y₂O₃ catalysts (both the deposited and ‘SA’ versions) is needed (and could assist with identifying the presence of SA Ni). At a minimum, assessing Ni reducibility, Ni dispersion and high resolution imaging of the catalysts should be performed.

Response: We thank for the reviewer’s good comments. To confirm the Ni state in SA Ni/Y₂O₃ nanosheets, we used the aberration-corrected TEM technique to identify the Ni distribution in SA Ni/Y₂O₃ sheets shown in Fig. 1a (Fig. 3a in revised

manuscript, see Page 17). The detailed description was shown in our revised manuscript as below in red color (see Page 7):

Fig. 1 Characterization of the Ni in SA Ni/Y₂O₃ nanosheets. (a) Aberration-corrected TEM image of SA Ni/Y₂O₃ nanosheets. (b) EXAFS spectra of the Ni K-edge of SA Ni/Y₂O₃ nanosheets, NiO and Ni foil. (c) Fourier transform (FT) of the Ni K-edge of SA Ni/Y₂O₃ nanosheets, NiO and Ni foil. (d) Schematic model of SA Ni/Y₂O₃ nanosheet, Ni (blue), Y (red), O (gray) and corresponding FT-EXAFS fitting curves for SA Ni/Y₂O₃ nanosheet. The scale bar in (a) is 2 nm.

“To identify the presence of Ni on SA Ni/Y₂O₃ sheets, the aberration-corrected TEM measurements were performed. Fig. 3a indicates that no precipitates large than 1 nm can be found in these nanosheets and only several dark tiny dots were dispersed on the whole nanosheets. It is noted that the diameter of these dots is 1-3 Å, similar to the size of single atoms or small clusters composed by several atoms.”

Further, we also added the TEM and HRTEM images of Ni nanoparticles decorated on Y_2O_3 nanosheets ($\text{Ni}/\text{Y}_2\text{O}_3$) in Fig. 2 (Supplementary Fig. 1 in revised Supplementary Materials, see Page 9) for comparison. The detailed description was shown in our revised Supplementary Materials as below in red color (see Page 9):

Fig. 2 (a, b) TEM and HRTEM images of $\text{Ni}/\text{Y}_2\text{O}_3$ nanosheets.

“The TEM image of $\text{Ni}/\text{Y}_2\text{O}_3$ nanosheets showed some nanoparticles distributed on the nanosheets (Supplementary Fig. 1a) and the HRTEM image revealed the crystalline nature of the nanoparticles with 0.218 nm of lattice spacing, corresponding to the (002) plane of metallic Ni,¹ thus, confirming that the crystalline precipitations on nanosheets are Ni nanoparticles.”

Measuring Ni dispersion would enable TOF to be determined and be a better comparison to illustrate the strong activity of SA $\text{Ni}/\text{Y}_2\text{O}_3$ (and its origin) relative to the other systems listed in Table S2.

Response: We are grateful to the reviewer for the valuable suggestion. We have added the TOF values of SA $\text{Ni}/\text{Y}_2\text{O}_3$ sheets and other catalysts in Table 1 (Supplementary Table 3 in revised Supplementary Materials, see Page 19), confirming the best CO_2 methanation of SA $\text{Ni}/\text{Y}_2\text{O}_3$ sheets compared with other noble metal free catalysts as far as we known. The detailed description was shown in our revised manuscript as below in red color (see Page 8, 9):

Table 1. The thermal CO_2 methanation performances of different catalysts.

Catalysts	Temperature (°C)	TOF (CO ₂)	References
SA Ni/Y ₂ O ₃	200	0.023	This work
Ni/Co ₃ O ₄	200	0.003	1
Co NP	200	0.005	2
Ni/VO _x	210	0.0023	3
Ni/ZrO ₂	235	0.058	4
Ni/Al ₂ O ₃	300	5.7	5
Ni/SiO ₂	300	1.61	6
Co/ZrO ₂	400	0.2	7
NiFe	300	5.9	8

“We compare some noble metal free catalysts for thermal CO₂ methanation in Supplementary Table 3. Remarkably, the thermal CO₂ methanation activity of SA Ni/Y₂O₃ nanosheets is higher than those reported Ni-based catalysts.”

The authors use 12hrs of continuous operation under simulated solar conditions to demonstrate catalyst stability and run it over one outdoor cycle to demonstrate stable performance which is insufficient to support the claim of a stable catalyst. The catalyst needs to be run for a much longer time-frame as well as be subject to multiple heating/cooling cycles (to replicate outdoor operation) before this claim can be made with conviction. Additionally, the catalyst should be characterized post-stability and post-cycling tests to ensure it has maintained its original integrity.

Response: Thank for the reviewer’s comments. We have prolonged the thermal CO₂ methanation test to 90 hours with several heating/cooling cycles as shown in Fig. 3c (Fig. 4c in revised manuscript, see Page 18). Further, we also showed the aberration-corrected TEM image and TEM image of SA Ni/Y₂O₃ nanosheets after 90

hours thermal CO₂ methanation test in Fig. 3d (Fig. 4d in revised manuscript, see Page 18) and Fig. 4 (Supplementary Fig. 6 in revised Supplementary Materials, see Page 14), respectively. The detailed description was shown in our revised manuscript as below:

Fig. 3 Thermocatalytic CO₂ hydrogenation experiments. (a) CO₂ conversion of SA Ni/Y₂O₃ nanosheets (SA Ni/Y₂O₃), Ni NPs/Y₂O₃ nanosheets (Ni/Y₂O₃) as a function of temperature. (b) CH₄ and CO yields in CO₂ hydrogenation over SA Ni/Y₂O₃ nanosheets as a function of temperature. (c) CO₂ hydrogenation versus reaction time over SA Ni/Y₂O₃ nanosheets at 240 °C. (d) The Aberration-corrected TEM image of SA Ni/Y₂O₃ nanosheets after stability test shown in Fig. 4c. Reaction condition: 100 mL min⁻¹ of reaction gas (2.5% CO₂ + 10 % H₂ + 87.5% N₂), 100 mg of catalysts. The scale bar in (d) is 2 nm.

Fig. 4 TEM image of SA Ni/Y₂O₃ nanosheets after 90 hours' heating/cooling CO₂ methanation test.

“During 90 hours' test heating/cooling test, the CO₂ conversion rate with the SA Ni/Y₂O₃ nanosheets maintains at ~87 % at 240 °C (Fig. 4c), evidencing the excellent catalytic stability. Supplementary Fig. 6 shows that the morphology of SA Ni/Y₂O₃ nanosheets is retained after 90 hours' test. And there are still several dark tiny dots observed on the SA Ni/Y₂O₃ nanosheets after stability test, rather than clear Ni precipitation (Fig. 4d), confirming the robustness of SA Ni/Y₂O₃ nanosheets in thermal CO₂ methanation.”

Moreover, we prolonged the photothermal CO₂ methanation test from 12 hours to 55 hours as shown in Fig. 5 (Supplementary Fig. 8 in revised Supplementary Materials, see Page 16) to evidence the robust of SA Ni/Y₂O₃ nanosheets in solar driven-CO₂ methanation.

Fig. 5 CO₂ hydrogenation over SA Ni/Y₂O₃ nanosheets under 1 kW m⁻² of simulated solar light irradiation in the photothermal system versus reaction time.

According to Figure 5b the SA Ni/Y₂O₃ catalyst gives a methane yield of around 90% with no CO produced. Does this mean that 10% of the carbon is unaccounted for or that the 90% yield is a reflection of the 90% CO₂ conversion reached by the system? If it is the later, then the result would be better presented as a selectivity % rather than a yield %. If it is the former, then what has happened to the other 10% of carbon?

Response: We are grateful to the reviewer for the valuable comments. The reason for non-100% CO₂ conversion is that the diameter of the reactor is too thick. As shown in the below Fig. 6a, the quartz tube we used in the manuscript is in 38 mm diameter, which is too thick to fill the reactor with catalyst, making the CO₂ can not be completely reacted with catalysts (Fig. 6c). When we used the thinner tube with 8 mm diameter (Fig. 6b), the catalysts can full fill the space of tube and realizing nearly 100% CO₂ conversion rate at 250 °C (Fig. 6c).

The coating of selective light absorber on quartz tube was made by Hangzhou Ruijia Precision Science Instrument Co., Ltd. Up to date, they can coat the selective light absorber on the thick quartz tube (38 mm diameter) rather than on the thinner one (8 mm). Therefore, we used the quartz tube with 38 mm diameter in both thermal and photothermal CO₂ methanation.

Fig. 6 (a, b) The schematic diagram of thermal CO₂ methanation reactor system with 38 and 8 mm diameter respectively. (c) the corresponding CO₂ conversion of 3.9 wt% SA Ni/Y₂O₃ nanosheets (SA Ni/Y₂O₃) as a function of temperature in the two reactors respectively (red line directed to the reaction in 38 mm diameter of tube, blue line directed to the reaction in 8 mm diameter of tube). Reaction condition: 100 mL min⁻¹ of reaction gas (2.5% CO₂ + 10 % H₂ + 87.5% N₂), 100 mg of catalysts.

The written English requires substantial improvement to remove the many grammatical errors and spelling mistakes (e.g. temperature, anchored, irradiation). Words such as ‘huge’ and ‘remarkable’ should be avoided and some of the terminology (e.g. ‘very few’, ‘is hard to be’, ‘nearly 100%’, ‘We listed some excellent catalysts’) should be revised to be more scientific. The sentence ‘In other words.....materials up to now’ doesn’t make sense.

Response: We thank the reviewer for pointing this out, and we have revised the whole manuscript to make it more satisfactory. The detailed description was shown in our revised manuscript in blue color.

Reviewer #2 (Remarks to the Author):

This manuscript reports CO₂ hydrogenation over Ni-based catalyst with heating of solar light. However, hydrogenation of CO₂ over Ni-based catalysts is widely

investigated, and it is a commercialized process. Furthermore, it is a common way to heat catalysts by solar light. Increase in heating efficiency by insulating a reactor is a conventional techniques.

Response: We are grateful to the reviewer for the valuable comments. Regarding the reviewer's comments, we would like to re-emphasize the novelty of this manuscript on increasing the temperature as below:

Metallic nanoparticles, carbon based materials, etc. have been widely investigated as photothermal catalysts due to their full sunlight absorption ability.^{9,10} Although they can full absorb sunlight, their thermal radiation is in the maximum degree too, due to their blackbody nature.^{11,12} The severe thermal radiation prevents the heat storage of photothermal materials, leading to 80 °C ceiling limit of photothermal materials under 1 solar irradiation,^{13,14} impossible to trigger the photothermal reactions. Selective light absorber is a kind of unique optical materials, which can not only fully absorb sunlight but also with little thermal radiation, highly conducive to thermal storage.¹⁵ Thus, we construct a new photothermal system using selective light absorber, which could heat catalysts to 288 °C temperature under weak sunlight irradiation ($1 \text{ kW}\cdot\text{m}^{-2}$), three times higher than that in traditional photothermal catalysis system, thus, for the first time realizing ambient sunlight-driven photothermal CO₂ methanation.

In addition, we made a comparison between the selective light absorber and typical photothermal materials with and without vacuum thermal insulating technology shown in Table 2 (Supplementary Table 1 in revised Supplementary materials, see Page 17). The detailed description was shown in our revised manuscript as below in red color (see Page 5, 6):

Table 2. The sunlight driven temperature of different materials in air and in vacuum respectively. The sunlight intensity is $1.0 \text{ kW}\cdot\text{m}^{-2}$.

Materials	CNT	Graphene	2wt% Au/Al ₂ O ₃	4wt% Ni/Y ₂ O ₃	CuO	2wt% Ru/Al ₂ O ₃	Selective light absorber
Temperature	70	79	70	78	82	64	260

in air (°C)							
Temperature	98	86	88	97	103	82	300
in vacuum (°C)							

“We have found that the selective light absorber can be heated up to 300 °C under one solar irradiation (Supplementary Table 1). We have compared the temperatures achieved with several typical photothermal materials under 1 solar irradiation, and 103 °C is the highest temperature of photothermal materials with vacuum thermal insulation (Supplementary Table 1), which is only 34% of the temperature realized by selective light absorber (300 °C). Therefore, the technique of selective light absorber could create unpredictable high temperature just by absorbing ambient sunlight.”

Furthermore, the characterization of catalytic active sites before and after reaction are little.

Response: We thank the reviewer for pointing this out, and we added a series of characterizations of catalytic active sites before and after reaction according to the reviewer’s suggestion. To confirm the Ni state in SA Ni/Y₂O₃ nanosheets, we used the aberration-corrected TEM technique to identify the Ni distribution in SA Ni/Y₂O₃ sheets shown in Fig. 1a (Fig. 3a in revised manuscript, see Page 17). We also added the FT-EXAFS fitting curve of our proposed model for SA Ni/Y₂O₃ in Fig. 1d (Fig. 3d in revised manuscript, see Page 17) and the detailed description was shown in our revised manuscript as below in red color (see Page 7, 8):

Fig. 1 Characterization of the Ni in SA Ni/Y₂O₃ nanosheets. (a) Aberration-corrected TEM image of SA Ni/Y₂O₃ nanosheets. (b) EXAFS spectra of the Ni K-edge of SA Ni/Y₂O₃ nanosheets, NiO and Ni foil. (c) Fourier transform (FT) of the Ni K-edge of SA Ni/Y₂O₃ nanosheets, NiO and Ni foil. (d) Schematic model of SA Ni/Y₂O₃ nanosheet, Ni (blue), Y (red), O (gray) and corresponding FT-EXAFS fitting curves for SA Ni/Y₂O₃ nanosheet. The scale bar in (a) is 2 nm.

“To identify the presence of Ni on SA Ni/Y₂O₃ sheets, the aberration-corrected TEM measurements were performed. Fig. 3a indicates that no precipitates large than 1 nm can be found in these nanosheets and only several dark tiny dots were dispersed on the whole nanosheets. It is noted that the diameter of these dots is 1-3 Å, similar to the size of single atoms or small clusters composed by several atoms.”

“The structure of amorphous Y₂O₃ decorated with single Ni atoms is illustrated in the inset of Fig. 3d. And the simulated Ni coordinated FT-EXAFS spectrum of this model (Fig. 3d) fits with the measured result of the SA Ni/Y₂O₃ nanosheets, further

confirming the dominant distribution of single Ni atoms on the SA Ni/Y₂O₃ nanosheets.”

Moreover, we have added the 90 hours thermal CO₂ methanation test with several heating/cooling cycles shown in Fig. 3c (Fig. 4c in revised manuscript, see Page 18), showed the aberration-corrected TEM image and TEM image of SA Ni/Y₂O₃ nanosheets after 90 hours thermal CO₂ methanation test in Fig. 3d (Fig. 4d in revised manuscript, see Page 18) and Fig. 4 (Supplementary Fig. 6 in revised Supplementary materials, see Page 14), respectively. The detailed description was shown in our revised manuscript as below (see Page 9):

Fig. 3 Thermocatalytic CO₂ hydrogenation experiments. (a) CO₂ conversion of SA Ni/Y₂O₃ nanosheets (SA Ni/Y₂O₃), Ni NPs/Y₂O₃ nanosheets (Ni/Y₂O₃) as a function of temperature. (b) CH₄ and CO yields in CO₂ hydrogenation over SA Ni/Y₂O₃ nanosheets as a function of temperature. (c) CO₂ hydrogenation versus reaction time over SA Ni/Y₂O₃ nanosheets at 240 °C. (d) The Aberration-corrected TEM image of SA Ni/Y₂O₃ nanosheets after stability test shown in Fig. 2c. Reaction condition: 100

mL min⁻¹ of reaction gas (2.5% CO₂+10 % H₂+ 87.5% N₂), 100 mg of catalysts. The scale bar in (d) is 2 nm.

Fig. 4 TEM image of SA Ni/Y₂O₃ nanosheets after 90 hours' heating/cooling CO₂ methanation test.

“During 90 hours' test heating/cooling test, the CO₂ conversion rate with the SA Ni/Y₂O₃ nanosheets maintains at ~87 % at 240 °C (Fig. 4c), evidencing the excellent catalytic stability. Supplementary Fig. 6 shows that the morphology of SA Ni/Y₂O₃ nanosheets is retained after 90 hours' test. And there are still several dark tiny dots observed on the SA Ni/Y₂O₃ nanosheets after stability test, rather than clear Ni precipitation (Fig. 4d), confirming the robustness of SA Ni/Y₂O₃ nanosheets in thermal CO₂ methanation.”

References

- 1 Wang, Y., Arandiyana, H., Scott, J., Dai, H. & Amal, R. Hierarchically Porous Network-Like Ni/Co₃O₄: Noble Metal-Free Catalysts for Carbon Dioxide Methanation. *Advanced Sustainable Systems* **2**, 1700119, doi:doi:10.1002/advsu.201700119 (2018).

- 2 Beaumont, S. K. *et al.* Combining in Situ NEXAFS Spectroscopy and CO₂ Methanation Kinetics To Study Pt and Co Nanoparticle Catalysts Reveals Key Insights into the Role of Platinum in Promoted Cobalt Catalysis. *Journal of the American Chemical Society* **136**, 9898-9901, doi:10.1021/ja505286j (2014).
- 3 Lu, X. *et al.* VOx promoted Ni catalysts supported on the modified bentonite for CO and CO₂ methanation. *Fuel Processing Technology* **135**, 34-46, doi:https://doi.org/10.1016/j.fuproc.2014.10.009 (2015).
- 4 Jia, X., Zhang, X., Rui, N., Hu, X. & Liu, C.-j. Structural effect of Ni/ZrO₂ catalyst on CO₂ methanation with enhanced activity. *Applied Catalysis B: Environmental* **244**, 159-169, doi:https://doi.org/10.1016/j.apcatb.2018.11.024 (2019).
- 5 He, L., Lin, Q., Liu, Y. & Huang, Y. Unique catalysis of Ni-Al hydrotalcite derived catalyst in CO₂ methanation: cooperative effect between Ni nanoparticles and a basic support. *Journal of Energy Chemistry* **23**, 587-592, doi:https://doi.org/10.1016/S2095-4956(14)60144-3 (2014).
- 6 Aziz, M. A. A. *et al.* Highly active Ni-promoted mesostructured silica nanoparticles for CO₂ methanation. *Applied Catalysis B: Environmental* **147**, 359-368, doi:https://doi.org/10.1016/j.apcatb.2013.09.015 (2014).
- 7 Li, W. *et al.* ZrO₂ support imparts superior activity and stability of Co catalysts for CO₂ methanation. *Applied Catalysis B: Environmental* **220**, 397-408, doi:https://doi.org/10.1016/j.apcatb.2017.08.048 (2018).
- 8 Pandey, D. & Deo, G. Effect of support on the catalytic activity of supported Ni-Fe catalysts for the CO₂ methanation reaction. *Journal of Industrial and Engineering Chemistry* **33**, 99-107, doi:https://doi.org/10.1016/j.jiec.2015.09.019 (2016).
- 9 Oara Neumann, A. S. U., Jared Day, Surbhi Lal, Peter Nordlander, and Naomi J. Halas. Solar Vapor Generation Enabled by Nanoparticles.pdf. *ACS nano* **7**, 42-49 (2013).
- 10 Bae, K. *et al.* Flexible thin-film black gold membranes with ultrabroadband plasmonic nanofocusing for efficient solar vapour generation. *Nature communications* **6**, 10103, doi:10.1038/ncomms10103 (2015).
- 11 Zeng, Y., Wang, K., Yao, J. & Wang, H. Hollow carbon beads for significant water evaporation enhancement. *Chemical Engineering Science* **116**, 704-709, doi:10.1016/j.ces.2014.05.057 (2014).
- 12 Xu, N. *et al.* Mushrooms as Efficient Solar Steam-Generation Devices. *Advanced materials* **29**, doi:10.1002/adma.201606762 (2017).
- 13 Wang, J. *et al.* High-Performance Photothermal Conversion of Narrow-Bandgap Ti₂O₃ Nanoparticles. *Advanced materials* **29**, doi:10.1002/adma.201603730 (2017).
- 14 Wang, Y., Zhang, L. & Wang, P. Self-Floating Carbon Nanotube Membrane on Macroporous Silica Substrate for Highly Efficient Solar-Driven Interfacial Water Evaporation. *ACS Sustainable Chemistry & Engineering* **4**, 1223-1230, doi:10.1021/acssuschemeng.5b01274 (2016).
- 15 Cao, F., McEnaney, K., Chen, G. & Ren, Z. A review of cermet-based spectrally selective solar absorbers. *Energy & Environmental Science* **7**, 1615-1627, doi:10.1039/C3EE43825B (2014).

At last, we wish to thank the Reviewers and the Editor again for the very constructive comments and suggestions to improve the quality of our manuscript. Thank you very much!

Reviewers' comments:

Reviewer #1 (Remarks to the Author):

The authors have addressed the bulk of the original comments to a satisfactory extent have performed most of the requested additional characterisation of the materials (pre and post reaction) to strengthen their claims, evaluated the material stability more thoroughly and provided additional evidence of their SAC Ni. Additional comments pertaining to the revised version are provided below.

Can the authors evaluate Ni reducibility on the SAC Ni and Ni nanoparticle (on Y2O3) systems as recommended in the original comments. Such evaluation will give further information and understanding on the difference in interaction between the Ni and the Y2O3 support in terms of SMSI.

The authors should take care when proofing their manuscript for spelling errors (e.g. 'precipitation' in the text relating to Figure 4). The grammar may have been checked but it still requires improvement. In the original comments it was recommended the authors avoid the use of subjective descriptive words such as 'remarkable' yet they use the word 'remarkably' when describing the activity comparison of their material to others (in relation to Table 3). Please remove and refrain from using such words to describe material performance.

Reviewer' comments

Reviewer #1 (Remarks to the Author):

The authors have addressed the bulk of the original comments to a satisfactory extent have performed most of the requested additional characterisation of the materials (pre and post reaction) to strengthen their claims, evaluated the material stability more thoroughly and provided additional evidence of their SAC Ni. Additional comments pertaining to the revised version are provided below.

Can the authors evaluate Ni reducibility on the SAC Ni and Ni nanoparticle (on Y₂O₃) systems as recommended in the original comments. Such evaluation will give further information and understanding on the difference in interaction between the Ni and the Y₂O₃ support in terms of SMSI.

Response: We are grateful for the reviewer's positive and insightful comments. According to the reviewer's comment, we have added the Ni 2P XPS of SA Ni/Y₂O₃ and Ni/Y₂O₃ nanosheets in Supplementary Fig. 5 to show the Ni reducibility conditions, and the detailed description was shown in our revised Supplementary Materials (see Page 13) and revised manuscript (see Page 8) in blue colour:

In revised Supplementary Materials

Supplementary Fig. 5 (a) The Ni 2p XPS spectra of SA Ni/Y₂O₃ nanosheets (red) and Ni/Y₂O₃ nanosheets (blue).

“The binding energy of Ni 2p_{3/2} is 853.1 eV for SA Ni/Y₂O₃ nanosheets, a little lower than that of NiO (853.7 eV),¹ revealing the oxidation state of Ni in SA Ni/Y₂O₃ nanosheets. The binding energy of Ni 2p_{3/2} is 852.0 eV for Ni/Y₂O₃ nanosheets, similar to the metallic Ni (852.0 eV),¹ confirming the metallic state of Ni in Ni/Y₂O₃ nanosheets. ”

In revised manuscript

“showing an oxidation state of Ni species, in comparison with the metallic state of Ni in Ni/Y₂O₃ nanosheets (Supplementary Fig. 5).”

The authors should take care when proofing their manuscript for spelling errors (e.g.

‘precipitation’ in the text relating to Figure 4). The grammar may have been checked but it still requires improvement. In the original comments it was recommended the authors avoid the use of subjective descriptive words such as ‘remarkable’ yet they use the word ‘remarkably’ when describing the activity comparison of their material to others (in relation to Table 3). Please remove and refrain from using such words to describe material performance.

Response: We are grateful to the reviewer for pointing this out. We have revised spelling error ‘precipitation’ as ‘precipitations’, and deleted the word ‘remarkably’. Further, we have revised the English of the whole manuscript. The detailed description was shown in our revised manuscript in red color.

At last, we wish to thank the Reviewer and the Editor again for the very constructive comments and suggestions to improve the quality of our manuscript. Thank you very much!

REVIEWERS' COMMENTS:

Reviewer #1 (Remarks to the Author):

The comments have been addressed and while there remain grammatical errors throughout the manuscript the authors have made an effort to improve this aspect.

REVIEWERS' COMMENTS:

Reviewer #1 (Remarks to the Author):

The comments have been addressed and while there remain grammatical errors throughout the manuscript the authors have made an effort to improve this aspect.

Response: We are grateful to the reviewer for pointing this out. We have improved the English of the whole manuscript by the English language editing service suggested by Nature Communications.

At last, we wish to thank the Reviewer and the Editor again for the very constructive comments and suggestions to improve the quality of our manuscript. Thank you very much!